# The Next-Generation Probiotic *E. coli 1917*-pSK18a-MT Ameliorates Cadmium-Induced Liver Injury by Surface Display of Metallothionein and Modulation of Gut Microbiota

**DOI:** 10.3390/nu16101468

**Published:** 2024-05-13

**Authors:** Yan Zhang, Hong Huang, Chuanlin Luo, Xinfeng Zhang, Yanjing Chen, Fenfang Yue, Bingqing Xie, Tingtao Chen, Changwei Zou

**Affiliations:** 1School of Resources and Environment, Nanchang University, Nanchang 330031, China; zhangy1718@163.com (Y.Z.); honghuang@ncu.edu.cn (H.H.); 2Queen Mary School, Jiangxi Medical College, Nanchang University, Nanchang 330031, China; 15680697626@163.com; 3School of Pharmacy, Jiangxi Medical College, Nanchang University, Nanchang 330031, China; zxf55623@sina.com; 4Department of Obstetrics & Gynecology, The 2nd Affiliated Hospital, Jiangxi Medical College, Nanchang University, Nanchang 330006, China; chenyanjing617@163.com; 5School of Life Sciences, Nanchang University, Nanchang 330031, China; yff9904@163.com; 6National Engineering Research Center for Bioengineering Drugs and the Technologies, Institution of Translational Medicine, Jiangxi Medical College, Nanchang University, Nanchang 330031, China; 7Department of Obstetrics & Gynecology, The 1st Affiliated Hospital, Jiangxi Medical College, Nanchang University, Nanchang 330006, China; xiebq_08@163.com

**Keywords:** Cadmium, *E. coli 1917*-pSK18a-MT, oxidative stress, inflammation, gut microbiota

## Abstract

Cadmium (Cd) is recognized as being linked to several liver diseases. Currently, due to the limited spectrum of drugs available for the treatment of Cd intoxication, developing and designing antidotes with superior detoxification capacity and revealing their underlying mechanisms remains a major challenge. Therefore, we developed the first next-generation probiotic *E. coli 1917*-pSK18a-MT that delivers metallothionein (MT) to overcome Cd-induced liver injury in C57BL/6 mice by utilizing bacterial surface display technology. The results demonstrate that *E. coli 1917*-pSK18a-MT could efficiently express MT without altering the growth and probiotic properties of the strain. Moreover, we found that *E. coli 1917*-pSK18a-MT ameliorated Cd contamination-induced hepatic steatosis, inflammatory cell infiltration, and liver fibrosis by decreasing the expression of aminotransferases along with inflammatory factors. Activation of the Nrf2-Keap1 signaling pathway also further illustrated the hepatoprotective effects of the engineered bacteria. Finally, we showed that *E. coli 1917*-pSK18a-MT improved the colonic barrier function impaired by Cd induction and ameliorated intestinal flora dysbiosis in Cd-poisoned mice by increasing the relative abundance of the Verrucomicrobiota. These data revealed that the combination of *E. coli 1917* and MT both alleviated Cd-induced liver injury to a greater extent and restored the integrity of colonic epithelial tissues and bacterial dysbiosis.

## 1. Introduction

Liver injury is a prevalent pathology featuring massive necrosis of hepatocytes, leading to a dramatic decline in liver function, and is responsible for more than 1 million deaths worldwide each year [1,2]. The disease, which is now associated with chemical pollutants, drugs, alcohol, and other factors, has developed into a critical public health problem on a regional and global scale. In humans and several animal models, cadmium (Cd) induces hepatotoxicity as a toxic substance in the environment [3]. According to studies, exposure to Cd can increase free radical production in the liver, disrupting the immune system and ultimately leading to hepatocellular carcinoma (HCC) due to inflammation, liver fibrosis and cirrhosis [4,5]. Liver damage and disease have now evolved into major threats to human health, attracting widespread attention in areas such as toxicology, public hygiene, nutrition, and food science.

Currently, in modern medicine, only symptomatic and non-specific metal ion chelators are available for Cd-poisoned patients, but their therapeutic effects are limited and they inevitably provoke adverse reactions [6]. In addition to this, endogenous protective mechanisms in biological organisms may also ameliorate the symptoms of Cd poisoning. Several studies have suggested that the nuclear factor erythroid-2-related factor (Nrf2)-signaling pathway is an essential defense system against oxidative stress damage, apoptosis, and inflammation, and is pivotal in neutralizing oxidative impairment [7]. Moreover, metallothionein (MT), an endogenous detoxification factor in organisms, reduces Cd toxicity by binding to Cd^2+^ through the sulfhydryl group of its intramolecular cysteine [8]. Thus, the induction of MT and activation of the Nrf2 signaling pathway may prevent the lethality and liver toxicity caused by Cd. However, the limited production of MT by the organism induced by Cd and the impossibility of avoiding Cd invasion by oral administration due to the presence of substantial amounts of proteases in the gastrointestinal tract prompted us to consider the Cd-resistant properties of exogenous MT in vivo [9,10]. Thus, the development of a secure and specific drug to target Cd-induced liver injury is imminent.

The intestinal microbiota and its metabolites are of increasing concern due to their involvement with modulating oxidative stress in tissues such as the liver [11]. Research has found that dysbiosis in gut ecology can cause damage to the intestinal mucosal barrier and produce leakage of translocated bacteria and gut-derived products [12,13], which can reach the liver through the portal system, leading to inflammation, oxidative stress, and liver disease [14]. Therefore, targeting the gut microbiota would be a potential therapeutic target to prevent or treat liver injury. Studies have demonstrated that probiotic supplementation can precede or attenuate hepatic and intestinal injuries by altering the structure of the gastrointestinal microbiota [15], which opens up the possibility of treating the disease of Cd-induced liver injury. For instance, a study reported that *Lactobacillus casei Shirota* was able to alter the gut microbiota and enhance the intestinal barrier function, thereby preventing liver and intestinal toxicity [16]. *E. coli Nissle 1917* (EcN), a promising probiotic for liver illnesses, is frequently used to modulate the intestinal microbiota and alleviate toxicant-induced inflammatory responses, among others [17]. Furthermore, EcN exhibits robust colonization properties and a well-defined genetic background [18], making it amenable to genetically engineered vectors. Study has shown that the use of EcN with surface-displaying peptides reduced Cd concentration in shrimp [19], which provides novel insights into reducing the accumulation of Cd in organisms. Our previous research also indicated that the plasmid-overexpressing engineered strain *E. coli 1917*-pet-28a-MT could effectively treat liver injury caused by subchronic Cd poisoning by suppressing the TLR4/NF-ĸB inflammatory signaling pathway [20].

Therefore, in this present work, we constructed the next-generation probiotic strain *E. coli 1917*-pSK18a-MT (EM) capable of surface expressing of MT. We evaluated the therapeutic efficacy of this strain against subchronic Cd-induced liver damage in mice and investigated its underlying mechanisms by hematoxylin eosin (HE) staining, Western blotting (WB), high-throughput sequencing, and other methods. The implementation of this study will provide a solid theoretical basis for the treatment of Cd-exposure-induced liver injury disease by engineered bacteria.

## 2. Materials and Methods

### 2.1. Construction and Evaluation of Next-Generation Probiotics In Vitro

The MT sequence (gene ID 856450) was synthesized by Bioengineering (Shanghai, China) Co., Ltd. and inserted into the pSK-18a plasmid after reverse transcription, which was introduced into *E. coli 1917* (E) to obtain the next-generation probiotic strain EM. Subsequently, its growth curves, plasmid stability, acid and bile salt resistance, and oxidation resistance were evaluated [21,22]. Ultimately, the expression of MT in EM was detected using a bacterial MT enzyme-linked immunosorbent assay (ELISA) kit (Shanghai Tongwei Industry Co., Ltd., Shanghai, China).

### 2.2. Design of Animal Experiments and Collection of Samples

Forty male C57BL/6 mice were used in this study and were purchased from Hunan SJA Laboratory Animal Co., Ltd., (Hunan, China). These mice were housed in a pathogen-free room at the Institute of Translational Medicine, Nanchang University (humidity 50 ± 10%, temperature 23 ± 2 °C, light-dark cycle 12:12), and were allowed to eat and drink freely. We randomly separated the mice into four groups after 1 week of adaptation: (i) C (intragastric administration of normal saline for 30 days); (ii) M (mice were treated with 100 mg/L CdCl_2_ (Aladdin, CN, Cat#C116342) aqueous solution for 30 days), (iii) E (mice were administered the probiotic bacterium E at 1 × 10^9^ cfu/(0.1 mL/10 g body weight) by gavage once a day for 30 days on the basis of drinking CdCl_2_ (Aladdin, CN, Cat#C116342) aqueous solution), (iv) EM (in addition to the CdCl_2_ aqueous treatment, EM was administered to mice by gavage at a dose of 1 × 10^9^ cfu/(0.1 mL/10 g body weight) daily for 30 days). We collected feces from the mice the day before the end of the experiment for further study. After the mice were fasted for 12 h, they were anesthetized with 2% isoflurane and executed. In order to obtain serum samples, the blood samples were placed at room temperature for half an hour and then centrifuged at 3000 rpm/min for 20 min and subsequently stored at −80 °C. Collected tissues, like liver and colon, were stored at −80 °C for future studies. The animal experiment was approved by the Experimental Animal Ethics Committee of Nanchang Leyou Biotechnology Co., Ltd., (Nanchang, China). (approval number: RyE2021070904).

### 2.3. Liver Index

The animals were weighed prior to sacrifice. The mice were anesthetized for dissection, and the liver tissues obtained were taken for weighing. The formula for calculating the liver index is as follows:Liver index (%) = liver weight (g)/body weight (g) × 100%

### 2.4. Histology and Histopathology

The liver and colon tissues of mice randomly selected from each group were fixed with 4% paraformaldehyde, paraffin-embedded and cut into 2~4 μm-thick serial sections, deparaffinized, stained with HE or sirius red, dehydrated, and cleaned, and then the HE and sirius-red-stained sections were observed under a light microscope.

### 2.5. Real-Time Fluorescence Quantitative PCR

The total RNA from liver tissues were extracted using Trizol reagent (Invitrogen, Waltham, MA, USA) according to the description of the relevant kit (Invitrogen, 15596026). The concentration of RNA was measured by a NanoDop spectrophotometer (Thermo Fisher Scientific, Cambridge, MA, USA). The genomic DNA was removed, and the reverse-transcribed cDNA samples were obtained after the reverse transcription of RNA using the Takara PrimeScript RT kit (Thermo Fisher Scientific, Cambridge, MA, USA). Gene expression was quantified using the 7900HT rapid real-time PCR system (ABI) and SYBR green fluorescent dye. The qPCR amplification program was set to start at 95 °C for 10 min, followed by annealing at 95 °C for 30 s, then annealing at 60 °C for 30 s and extending at 72 °C for 30 s for a total of 40 cycles. Finally, GAPDH was selected as the internal reference gene and analyzed using the 2^−ΔΔCt^ method. The primer sequences used in the experiment are given in Appendix A.

### 2.6. Measurement of Cd in Feces and Liver Tissue

0.1 g of each group of mice feces and liver tissue samples were added to 1 mL of PBS for homogenization, and 3 mL of concentrated nitric acid was added and left overnight, after which the samples were transferred to a polytetrafluoroethylene ablation tube to be ablated at 80 °C for 1 h. At the end of the ablution, 2 mL of H_2_O_2_ solution was added to continue to heat up to 120 °C until it turned into a colorless and transparent liquid, and then it was cooled down to below 60 °C, fixed, and mixed using ultrapure water. The Cd content in the samples was quantified by inductively coupled plasma emission spectrometry (ICP-MS) (Thermo Fisher Scientific lnc., Cambridge, MA, USA) and expressed as mg/g sample wet weight [20].

### 2.7. Western Blotting

Next, we weighed 0.1 g of liver in a centrifuge tube, added 1 mL of RIPA lysis buffer and 10 μL of protease inhibitor and phosphatase inhibitor, and placed it on ice for homogenization. The protein concentration in the supernatant was analyzed by the BCA method. Using 8–15% sodium dodecyl sulfate-polyacrylamide gel electrophoresis (SDS-PAGE), equal amounts of protein samples were electrophoresed and transferred to PVDF membranes (Millipore, Boston, MA, USA, IPVH00010). After being closed for 1 h at room temperature using skimmed milk, the membranes were incubated with their corresponding primary antibodies at 4 °C overnight (Appendix A). The membrane was then incubated with the secondary antibody for 1 h at room temperature. Finally, the intensity of the bands was displayed by chemiluminescence solution (Thermo Fisher, Waltham, MA, USA, 32209) in a gel imaging system. Measurements were made with ImageJ software (Version 1.52a) for subsequent analysis.

### 2.8. Determination of Oxidation-Related Factors in Liver and Serum

The levels of catalase (CAT) (A007-1-1), superoxide dismutase (SOD) (A001-1-1), and the malondialdehyde (MDA) (A003-1-1) in liver tissue, as well as the serum levels of alanine aminotransferase (ALT) (C009-2-1), aspartate aminotransferase (AST) (C010-2-1), alkaline phosphatase (ALP) (A059-2-1), and lactate dehydrogenase (LDH) (A020-2-1) were determined using commercial kits (Nanjing Jiancheng Institute of Bioengineering Research, Nanjing, China) according to the manufacturer’s description.

### 2.9. 16S rRNA Gene Sequencing

Microbial genomic DNA was extracted using a DNA extraction kit (Tiangen, Beijing, China, DP712) according to the manufacturer’s instructions and tested for concentration and purity using a spectrophotometer. To sequence the colony DNA fragments, the 16S rRNA V4 fragment was amplified with bacterial universal primers and bipartite sequenced using the Illumina platform (San Diego, CA, USA). Finally, the species composition was analyzed, as well as the species diversity and species differences.

### 2.10. Statistical Analysis

The analysis of the data was performed using Prism version 8.0.2 (GraphPad software, San Diego, CA, USA). The data were analyzed using one-way analysis of variance (ANOVA), followed by Tukey’s multiple comparison test. All data were expressed as mean ± standard deviation (SD). *p* < 0.05 and *p* < 0.01 were considered statistically significant.

## 3. Results

### 3.1. Evaluation of the Probiotic Properties of EM In Vitro

First, we investigated the growth of E and EM by turbidimetric assay within 24 h. The MT gene exhibited no significant effect on the growth of the bacteria as compared to the wild strain E. Additionally, we also found that both strains entered a stable growth period after 14 h (Figure 1A). Subsequently, to determine the expression of MT, quantitative analysis was performed using an ELISA kit. In Figure 1B, the MT content in the EM supernatant was 166 ± 6 pg/mL after 24 h of treatment under the same experimental conditions. Next, we evaluated the plasmid resistance of EM and showed that the engineered bacterium EM remained stable by reaching 89.31% after 25 days of passaging (Figure 1C). After that, a series of characterization experiments were performed on EM, and the results indicated that both E and EM had the ability to grow and survive under high degrees of acid and bile salts, which meant that these strains were able to tolerate conditions mimicking the gastrointestinal environmental (Figure 1D,E). Finally, the antioxidant properties of E and EM were evaluated, and we noticed that the scavenging ability of EM was significantly better than that of E, which indicated that EM had good antioxidant capacity (*p* < 0.05) (Figure 1F). The above results indicated that the constructed next-generation probiotic strain EM could efficiently express MT proteins without altering the growth and beneficial properties of the bacteria.

### 3.2. EM Treatment Reversed the Pathologic Changes in Cd-Intoxicated Mice

To explore the therapeutic effects of EM, we constructed a mouse model of Cd intoxication (Figure 2A) and recorded the body weight of the mice before sacrifice. Weight loss is an external symptom of Cd poisoning. As demonstrated in Figure 2B–D, the mice in the model group showed obvious weight loss after Cd treatment (C Vs M, 24.900 ± 1.200 Vs 22.8650 ± 0.991, *p* < 0.01), which was accompanied by liver enlargement and a remarkable increase in the proportion of liver to body weight (liver index) (C Vs M, 4.528 ± 0.539 Vs 5.771 ± 0.367, *p* < 0.01). After administration of EM, all the above symptoms were reduced in the mice, while the administration of E did not achieve a promising therapeutic effect. To further evaluate the residual Cd in the mice, we measured their feces and liver by ICP-MS. It was observed that the EM intervention greatly reduced the Cd levels in the liver (EM Vs M, 4.314 ± 0.373 Vs 5.652 ± 0.278, *p* < 0.01) and promoted the excretion of Cd in the organism (EM Vs M, 75.006 ± 4.365 Vs 53.006 ± 5.533, *p* < 0.01) (Figure 2E,F). In addition, the histopathological results indicated that the livers of the mice in group M were significantly damaged after Cd exposure, as manifested by disorganized hepatic cord arrangement, increased hepatic steatosis, and inflammatory cell infiltration (Figure 2G). Meanwhile, the sirius red results demonstrated that yellow reticulation was clearly observed in group M in contrast to group C, indicating Cd-induced collagen fiber production in the liver (Figure 2H). Notably, all the above phenomena were alleviated after EM treatment. Further, we found that the hepatic fibrosis protein α-SMA was considerably elevated after Cd exposure as compared to group C (C Vs M, 0.338 ± 0.038 Vs 1.393 ± 0.067, *p* < 0.01), but the expression level of α-SMA was reduced after EM administration (EM Vs M, 0.425 ± 0.054 Vs 1.393 ± 0.067, *p* < 0.01) (Figure 2I,J). These results suggest that EM administration significantly attenuated the pathologic changes in the organism caused by Cd poisoning.

### 3.3. Enhancement of Hepatic Dysfunction and Oxidative Stress Induced by Cd Intoxication by EM Administration

To evaluate the underlying mechanism by which EM alleviates Cd-induced hepatic damage in mice, we investigated the redox status in the liver as well as in the serum of Cd-contaminated mice. Relevant findings had demonstrated that the levels of AST, ALT, ALP, and LDH were widely used to assess hepatic impairment. Therefore, we used the changes in AST, ALT, ALP, and LDH to assess whether the liver function was normal after Cd exposure. The results revealed that Cd significantly increased serum AST (C Vs M, 54.541 ± 6.087 Vs 109.758 ± 8.945, *p* < 0.01), ALT (C Vs M, 47.731 ± 11.235 Vs 136.017 ± 7.244, *p* < 0.01), ALP (C Vs M, 43.509 ± 2.965 Vs 73.707 ± 4.677, *p* < 0.01), and LDH (C Vs M, 1232.227 ± 128.225 Vs 3109.005 ± 231.598, *p* < 0.01) activities compared with group C (Figure 3A–D), which confirmed the successful modeling of Cd-induced hepatotoxicity. The above significantly inhibited the elevation of marker enzyme activities in serum after EM administration. The levels of several liver-related oxidative stress markers were also assessed. The results demonstrated that MDA levels were significantly elevated in the livers of mice after Cd exposure (C Vs M, 3.886 ± 0.244 Vs 6.399 ± 0.254, *p* < 0.01), whereas the activities of CAT (C Vs M, 42.233 ± 1.795 Vs 17.870 ± 2.025, *p* < 0.01) and SOD (C Vs M, 253.475 ± 7.535 Vs 185.085 ± 9.641, *p* < 0.01) were significantly decreased. However, EM treatment dramatically reversed the increase in MDA (EM Vs M, 4.565 ± 0.206 Vs 6.399 ± 0.254, *p* < 0.01) content and restored SOD (EM Vs M, 225.728 ± 8.389 Vs 185.085 ± 9.641, *p* < 0.01) and CAT (EM Vs M, 31.880 ± 2.820 Vs 17.870 ± 2.025, *p* < 0.01) activities in the livers of Cd-contaminated mice (Figure 3E–G). Furthermore, WB results showed that Nrf2 (C Vs M, 1.232 ± 0.029 Vs 0.642 ± 0.058, *p* < 0.01) protein levels were reduced, as well as the Kelch-like ECH-associated protein 1 (Keap1) (C Vs M, 0.473 ± 0.029 Vs 1.162 ± 0.055, *p* < 0.01), and the protein expression levels were elevated in hepatic tissues after Cd exposure compared to group C. The administration of EM attenuated these changes induced by Cd (Figure 3H–J). These data provided strong evidence that EM could restore oxidative stress and hepatic dysfunction induced by Cd intoxication.

### 3.4. EM Treatment Reduces Inflammation Levels in the Liver Tissues of Cd-Exposed Mice

Considering the Cd-induced hepatocellular injury and the evident inflammatory cell infiltration in the liver, we detected the levels of pro-inflammatory factors including interleukin-6 (IL-6), interleukin-1β (IL-1β), and tumor necrosis factor-α (TNF-α) in the homogenates of the liver tissues of each group by q-PCR. The expression levels of IL-1β (C Vs M, 1.001 ± 0.076 Vs 1.950 ± 0.148, *p* < 0.01), IL-6 (C Vs M, 1.004 ± 0.117 Vs 1.647 ± 0.080, *p* < 0.01), and TNF-α (C Vs M, 1.003 ± 0.095 Vs 1.835 ± 0.143, *p* < 0.01) were elevated in the Cd-treated mice as compared to group C, indicating that Cd exposure activated the hepatic inflammatory responses but effectively reduced the expression of these pro-inflammatory factors after EM administration (Figure 4A–C). Moreover, phosphorylation of the p65 (p-p65) subunit exerts a pivotal position in the regulation of the NF-κB activation. Thus, we examined the expression level of p-p65 protein by WB. The results showed that EM intervention attenuated the Cd-induced p-p65 protein expression (EM Vs M, 0.796 ± 0.038 Vs 1.121 ± 0.046, *p* < 0.01) (Figure 4D,E). Taken together, these figures demonstrate that EM could successfully inhibit liver inflammation and the activation of p-p65 protein induced by Cd exposure.

### 3.5. EM Maintained the Intestinal Barrier Function and Reversed the Dysbiosis of Gut Microbiota in Cd-Intoxicated Mice

Cd exposure may disturb the integrity of the intestinal barrier and lead to intestinal dysbiosis. To determine whether EM influenced the intestinal homeostasis in Cd-poisoned mice, we histologically examined colonic tissues and measured the expression of tight junction proteins. HE staining showed significant inflammatory infiltration and intestinal epithelial damage in colonic tissues of group M as compared to group C, altering the morphological structure of the intestinal tract to a certain extent and showing significant lesions, which was interestingly attenuated more dramatically by EM treatment than in group E (Figure 5A). In addition, the expression of ZO-1 and Occludin proteins in the colon were detected using WB. As shown in Figure 5B–D, a significant reduction in these two prominent tight junction proteins’ activity in Cd-exposed mice was observed as compared to group C. However, EM treatment markedly enhanced these proteins’ expression levels. These results demonstrated that the administration of EM could maintain intestinal epithelial integrity in Cd-exposed mice, thereby enhancing the intestinal barrier function.

With the help of high-throughput sequencing, we analyzed the modifications in the intestinal microbiota of the mice that occurred as a result of EM exposure. The outcomes demonstrated that the α-diversity of group M was greatly reduced in all indices, such as Chao1 and the Faith_pd index, as compared to group C (Figure 5E,F). The Venn results revealed that there were a total of 189 common OTUs in all groups, and the numbers of the unique OTUs were 479, 202, 301, and 339 in the C, M, E, and EM groups, respectively (Figure 5G). The principal coordinate analysis (PCoA) showed that the samples in groups M and E were significantly different from the other groups (Figure 5H). In addition, at the phylum level, Bacteroidota, Bacillota, Verrucomicrobiota, and Pseudomonadota were the most prevalent and dominant phyla in these four groups, occupying 96.67%, 94.3%, 94.7%, and 96.03% of the total number of sequences of the four taxa, respectively (Figure 5I). At the genus level, a reduction in the relative abundance of *Akkermansia* was found in the Cd-poisoned mice (M Vs C, 5.27% Vs 28.95%, *p* < 0.01). Nevertheless, EM treatment reversed this trend (Figure 5J,K). These data showed that the composition of the gut community could be affected by Cd exposure, and that administration of EM improved the gut microbiota of Cd-poisoned mice.

## 4. Discussion

Cd is a ubiquitous environmental contaminant that can promote an increased mortality risk from liver disease with long-term contact [23]. Today, scientists still have not developed specific therapeutic drugs for Cd poisoning. In a previous study, our team found that plasmid overexpression-type engineered bacterium *E. coli 1917*-pet-28a-MT could ameliorate Cd-induced liver injury [20]. However, the intracellular expression of this strain suffers from the disadvantages that the plasmid is susceptible to degradation by intracellular enzymes, which can become toxic to the cells by accumulating in them. With a view to overcoming these limitations of the strain, we proposed to construct a next-generation probiotic strain, EM, using surface display technology. As we all know, *E. coli Nissle 1917* (EcN) is an enteric commensal bacterium [24] that is an ideal carrier for in situ synthesis of therapeutic molecules in the intestine because of its potential to deliver biologically active molecules in vivo as well as its excellent colonization characteristics [18,25,26]. In conclusion, the application of surface-display technology could enable EM to better stabilize its ability to target Cd in the human intestine.

As mentioned above, the next-generation probiotic strain constructed in this study could express MT proteins efficiently without altering the bacterial growth and had good stability. The acid and bile salt tolerance tests demonstrated that the strain was able to tolerate low pH and a high concentration of bile hydrochloric acid, and it also had the abilities to survive in the intestinal tract and exhibit probiotic efficacy. In addition, the results of the antioxidant-resistance experiments showed that EM had better scavenging ability than E for free radicals such as DPPH, ·O_2_-, and ·OH, indicating that EM had better antioxidant ability. The above findings demonstrated that EM possessed the ability to be orally administered and was resistant to gastric acid (Figure 1).

Following the promising results of earlier studies, we used a Cd-intoxicated mouse model to explore the clinical effects and the potential mechanism of the effects of EM. It was found that changes in body weight and organ mass are usually used as indicators of body health status and organization toxicity [27,28]. The present study revealed that mice exposed to Cd significantly lost body weight, accompanied by a marked increase in liver coefficients. However, the health statuses of the mice were modified better after supplementation with *E. coli 1917*-pSK18a-MT as compared to the administration of the wild-type strain E. Moreover, further analysis revealed that Cd exposure increased hepatic Cd accumulation, whereas EM treatment significantly reduced hepatic Cd content, which might be attributed to the chelation of MT proteins expressed by EM with Cd, which promoted the exocytosis of Cd and, consequently, led to the reduction of Cd levels in the liver tissues. This conjecture was reinforced by the determination of Cd concentrations in feces. Histopathologic changes had been reported to be a direct parameter for assessing organ damage [29]. In this study, based on HE staining, Cd exposure caused liver pathology in mice, which was manifested by hemorrhaging [30], inflammatory cell infiltration [31], and hepatocyte swelling [32], similar to the findings of earlier reports [33]. However, it is noteworthy that the damaged liver tissues were improved by administration of E and EM, with EM showing the most optimal therapeutic effect. In addition, previous studies have demonstrated that Cd exposure increased the formation and deposition of collagen in liver tissues [34]. In this present study, the results of sirius red staining and α-SMA assay indicated that Cd exposure significantly enhanced α-SMA protein levels, and the administration of EM reversed these Cd-induced changes. These findings were consistent with previous reports [35], suggesting that EM treatment alleviated weight loss and liver tissue damage in mice, promoted the fecal excretion of Cd, and reduced the body’s uptake of Cd, which in turn reduced the production of liver fibrosis (Figure 2).

Some evidence suggests that Cd exposure disrupted the dynamic equilibrium of the cellular redox state, which in turn adversely affects liver function, as reflected by alterations in liver enzyme activities [36]. In line with these studies, our studies showed that chronic Cd exposure resulted in the remarkably enhanced activities of serums AST, ALT, ALP, and LDH [37]. The above results indicate that Cd treatment increased the cell membrane permeability, which in turn led to the disruption of cellular enzyme systems [38]. Furthermore, our data showed that exposure to Cd decreased SOD and CAT levels and increased the MDA contents in mice livers, which is consistent with previous reports [36]. As the main oxidation product of lipid peroxidation, MDA’s content could represent how much peroxidation has occurred within liver cell membranes [39], whereas SOD and CAT are two key antioxidant enzymes in the antioxidant defense system, controlling the levels of various reactive oxygen species (ROS) in the organism [40]. In this study, EM significantly reduced MDA formation in the liver; lowered the serum concentrations of AST, ALT, ALP, and LDH; and remarkably restored the functions of the antioxidant enzymes SOD and CAT. This indicated that EM was effective in lowering oxidative stress and augmenting cellular antioxidant enzymes activities in the livers of Cd-exposed mice. Furthermore, Cd exposure could counteract liver injury by activating various antioxidant pathways in addition to the above antioxidant enzymes. Additionally, Cd exposure could counteract liver injury by activating various antioxidant pathways in addition to the above antioxidant enzymes [41]. Among them, the Nrf2-Keap1 signaling pathway, as an important defense system against oxidative damage, is an essential target for the prevention of Cd-induced hepatic injury [42]. Consistent with prior reports, our work also revealed that the Nrf2 expression was markedly suppressed and the Keap1 expression was significantly elevated after Cd exposure [43], which was reversed by EM administration. Our findings demonstrated that EM could further inhibit liver injury in the Cd-exposed mouse model by modulating the Nrf2-Keap1 pathway (Figure 3).

Moreover, except for oxidative stress, inflammation as a secondary response serves a pivotal function in the pathogenesis of Cd-induced liver injury [23]. Previous research has identified the fact that Cd intoxication enhanced the release of pro-inflammatory cytokines by promoting the overproduction of ROS in the liver [44]. In this work, the expression of TNF-α, IL-1β, and IL-6 was markedly upregulated in the liver tissues of mice treated with Cd. In addition, the WB results showed an elevated expression of p-p65 in a Cd-induced hepatic injury. NF-ĸB is a redox-sensitive transcription factor that participates in the regulation of immunity, inflammation, cell proliferation, and other physiological processes. Once activated, it translocates to the nucleus to initiate the transcription of a variety of target genes [45]. Consistent with our data, Zou et al. confirmed that the expression of p-p65 was upregulated in the liver tissues of CdCl_2_-injected mice [20]. In addition, we speculated that EM possessed anti-inflammatory properties by downregulating the expression of TNF-α, IL-1β, IL-6, and p-p65 in mice livers. These findings were similar to those of previous studies [46]. Thus, based on these results, the NF-kB signaling pathway may contribute to the ameliorative effect of EM in Cd-induced liver injury by inhibiting inflammatory mediator production and the inflammatory response (Figure 4). Imbalance of intestinal bacteria has been reported to be well associated with the pathogenesis of Cd-exposure-induced liver injury [47]. Meanwhile, sustained Cd exposure leads to damage of intestinal epithelial cells, which in turn disrupts intestinal integrity [48]. Tight junction proteins (ZO-1, Occludin) are pivotal markers of the intestinal barrier, which perform essential functions in maintaining epithelial integrity and protecting the body from invasion by toxic and harmful substances [46,47]. In this work, the HE staining results showed that the intervention of EM effectively improved the pathological characteristics of mouse colon tissues. In addition, EM treatment upregulated the expression of intestinal permeability proteins ZO-1 and Occludin, thus enhancing the intestinal barrier function. Interestingly, treatment with wild-type strain E alone was weakly effective in reversing the expression of the above two proteins. Previous studies have confirmed that Cd exposure could alter the diversity and composition of the intestinal flora, which is compatible with the outcomes of this present study [49,50]. In this work, the intervention of EM slightly restored the α-diversity and β-diversity of the intestinal flora of Cd-intoxicated mice. Venn diagrams also showed that EM recovered the microbial diversity to some degree. In addition, the relative abundances of the Firmicutes and Verrucomicrobia were also changed after EM treatment. These results indicate that the next-generation probiotic EM could ameliorate Cd-exposure-induced liver injury by restoring intestinal permeability and modulating the relative abundance of the associated flora (Figure 5).

## 5. Conclusions

In summary, our findings demonstrate that the next-generation probiotic strain EM mediated Cd-exposure-induced oxidative stress and inflammatory responses through the surface display of MT. Meanwhile, EM also repaired the intestinal barrier function and reduced the entry of potentially toxic and harmful substances into the enterohepatic circulation, proving to be an excellent candidate for therapy of Cd exposure-induced liver injury (Figure 6). However, this study only explored the mechanism of the action of EM on foodborne Cd-exposure-induced liver injury in the healthy male mouse model, as well as the uncertainty of the expression of engineered bacteria in affecting gut-associated metabolites in the mice, which had a non-negligible impact on the conclusions that we drew.

## Figures and Tables

**Figure 1 nutrients-16-01468-f001:**
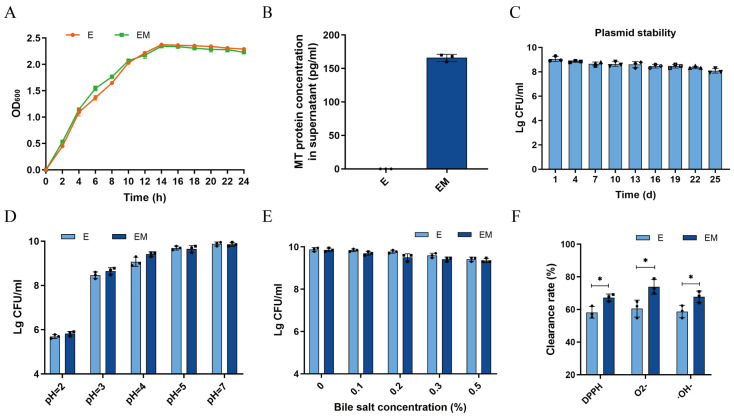
Evaluation of the microbial characterization of EM *in vitro*. (**A**) Growth curves of E and EM (*n* = 3); (**B**) expression of MT in EM detected by ELISA (*n* = 3); (**C**) EM plasmid stability assay (*n* = 3); (**D**) acid tolerance of E and EM (*n* = 3); (**E**) bile salt tolerance of E and EM (*n* = 3); and (**F**) antioxidant properties of E and EM (*n* = 3). E, *E. coli 1917*; EM, *E. coli 1917*-pSK18a-MT; MT, metallothionein; h, hour; d, day. * *p* < 0.05.

**Figure 2 nutrients-16-01468-f002:**
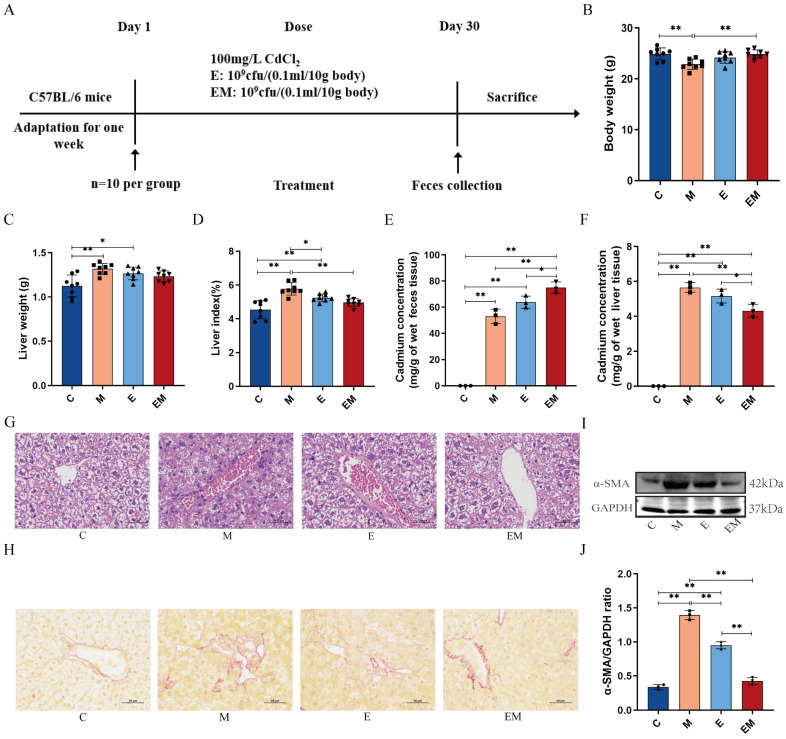
EM treatment reversed the pathologic changes in Cd-intoxicated mice. (**A**) Animal experimental design; (**B**) body weight (*n* = 8); (**C**) liver weight (*n* = 8); (**D**) liver index (*n* = 8); (**E**) measurement of Cd content in feces using ICP-MS (*n* = 3); (**F**) determination of Cd content in liver by ICP-MS (*n* = 3); (**G**) histopathological changes of liver observed by HE staining (scale bar = 50 μm); (**H**) sirius-red-stained images of liver tissue (scale bar = 50 μm); (**I**) WB analysis of α-SMA and GAPDH in liver; and (**J**) quantification of α-SMA by ImageJ based on internal reference GAPDH (*n* = 3). * *p* < 0.05 and ** *p* < 0.01.

**Figure 3 nutrients-16-01468-f003:**
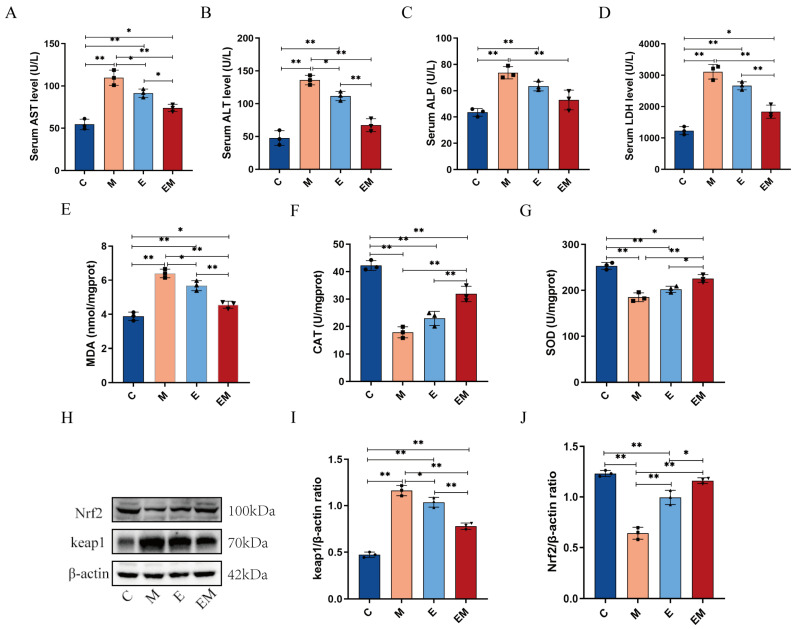
Enhancement of hepatic dysfunction and oxidative stress induced by Cd intoxication by EM administration. (**A**–**D**) serum levels of oxidative stress factors AST, ALT, ALP, and LDH (*n* = 3); (**E**–**G**) activities of MDA, CAT, and SOD in the liver (*n* = 3); (**H**) WB of Nrf2, Keap1, and β-actin in the liver; (**I**) quantification of Keap1 by ImageJ based on β-actin (*n* = 3); and (**J**) quantification of Nrf2 by ImageJ based on β-actin (*n* = 3). * *p* < 0.05 and ** *p* < 0.01.

**Figure 4 nutrients-16-01468-f004:**
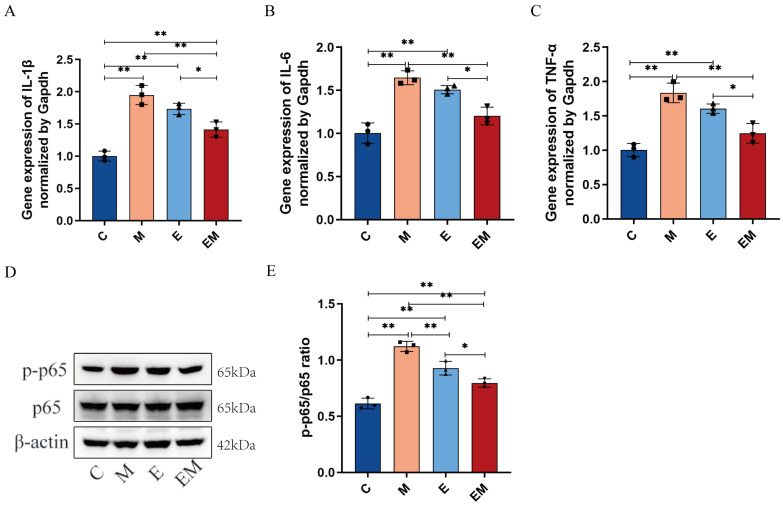
EM treatment reduces inflammation levels in liver tissue of Cd-exposed mice. (**A**–**C**) Relative mRNA expression levels of IL-1β, IL-6, and TNF-α in hepatic tissues detected by q-PCR (*n* = 3); (**D**) WB of p65, p-p65, and β-actin in liver; and (**E**) relative expression of p-p65/p65 by ImageJ (*n* = 3). * *p* < 0.05 and ** *p* < 0.01.

**Figure 5 nutrients-16-01468-f005:**
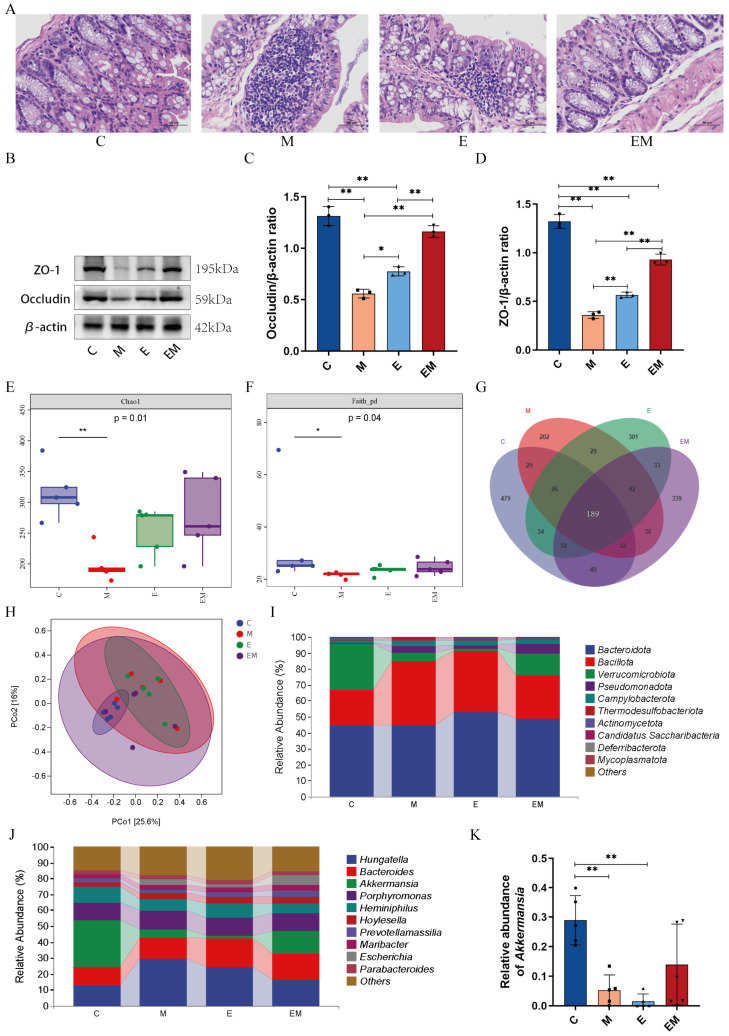
EM maintained the intestinal barrier function and reversed dysbiosis of intestinal flora in Cd-intoxicated mice. (**A**) HE-stained images of colonic tissues (scale bar = 50 μm); (**B**) WB of ZO-1, Occludin, and β-actin in the colon; (**C**,**D**) quantification of ZO-1, Occludin by ImageJ based on β-actin (*n* = 3); (**E**) Chao1 index (*n* = 5); (**F**) Faith_pd index (*n* = 5); (**G**) Venn map of OTUs (*n* = 5); (**H**) PCoA analysis (*n* = 5); (**I**) composition of microbiota at the phylum level (*n* = 5); (**J**) composition of microbiota at the genus level (*n* = 5); and (**K**) *Akkermansia* relative abundance (*n* = 5). * *p* < 0.05 and ** *p* < 0.01.

**Figure 6 nutrients-16-01468-f006:**
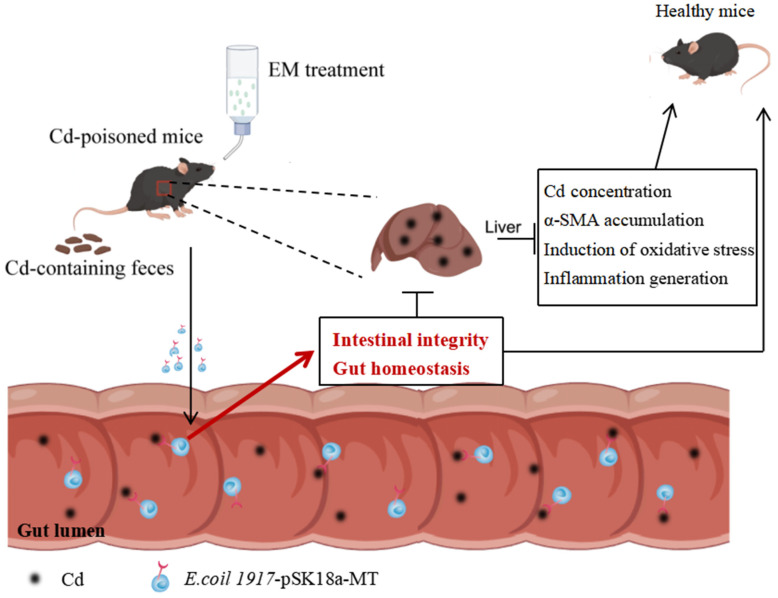
The underlying mechanism of the therapeutic effect of EM.

## Data Availability

The 16S rRNA dataset presented in this study was found in an online repository. The names of the repository/repositories and accession number(s) can be found below: NCBI Sequence Read Archive, BioProject ID PRJNA1068325. Other data will be made available by the authors without undue reservation.

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
