# Peer review of "The Next-Generation Probiotic E. coli 1917-pSK18a-MT Ameliorates Cadmium-Induced Liver Injury by Surface Display of Metallothionein and Modulation of Gut Microbiota"

_nutrients, 2024, doi:10.3390/nu16101468_

Round 1
Reviewer 1 Report
Comments and Suggestions for Authors
In this study outhors found that the probiotic E. coli 1917-pSK18a-MT could efficiently express metallothionein (MT) and that it ameliorated Cadmium contamination-induced hepatic steatosis, inflammatory cell infiltration, and liver fibrosis by decreasing the expression of aminotransferases along with inflammatory factors. The hepatoprotective effects of the engineered bacteria was also demonstrated by the activation of the Nrf2-Keap1 signaling pathway.
the mauscript is interesting and generally well written. The materials and methods provided are sufficient to replicate the experiments. Figures and tables are clear and easy to read.
my comments are listed below
Introduction: since Nrf2/keap1 signaling has an important role in this manuscript. The functions of this signaling deserve to be mentioned. In fact, it is involved in the onset and progression of sevaral diseases (as also recently reviewed PMID: 37296665,PMID: 38634043).
2.8. Determination of Oxidation-Related Factors in Liver and Serum:please add the product codes of the kits used
Figure 2G and H: please add a higher magnification to evaluate tissue morphology
Figure 5A: please add a higher magnification to evaluate tissue morphology
western blot images: please add the molecular weights in western blot images
please revise the punctuation and typing errors throughout the manuscript
Figure 6: remove the error red lines
the abbreviations must be written in full length the first time that they are mentioned
Author Response
Dear reviewer,
Thank you for reviewing our manuscript “The Next-Generation Probiotic E. coli 1917-pSK18a-MT Ameliorates Cadmium-Induced Liver Injury by Surface Display of Metallothionein and Modulation of Gut Microbiota” (Manuscript ID: nutrients-2976939), and we appreciate the time and effort that you dedicated to providing feedback on our manuscript and are grateful for the insightful comments on and valuable improvements to our paper. The manuscript was carefully revised and point-by-point response was listed below. We hope that your comments have been addressed accurately. And those changes are highlighted in the manuscript by the revision pattern as well as in underline/red.
Reviewer Comments:
Comment 1: Introduction: since Nrf2/keap1 signaling has an important role in this manuscript. The functions of this signaling deserve to be mentioned. In fact, it is involved in the onset and progression of sevaral diseases (as also recently reviewed PMID: 37296665,PMID: 38634043).
Response 1: Thanks for your suggestion. We apologize for not describing the function of the Nrf2 signaling pathway in the introduction of the article. To compensate for this omission, we have added a description of it in the article as follows:
Lines 53-56: “Several studies have suggested that the nuclear factor erythroid 2-related factor (Nrf2) signaling pathway is an essential defense system against oxidative stress damage, apoptosis, and inflammation, and is pivotal in neutralizing oxidative impairment”.
We hope that our correction will be recognized by you.
Comment 2: 2.8. Determination of Oxidation-Related Factors in Liver and Serum: please add the product codes of the kits used.
Response 2: We appreciate the reviewer’s comment. We apologize for our unclear description. In the revised manuscript, we rephased the description of Method section as follows,
In lines 167-171, “The levels of catalase (CAT) (A007-1-1), superoxide dismutase (SOD) (A001-1-1), malondialdehyde (MDA) (A003-1-1) in liver tissue, as well as the serum levels of alanine aminotransferase (ALT) (C009-2-1), aspartate aminotransferase (AST) (C010-2-1), alkaline phosphatase (ALP) (A059-2-1) and lactate dehydrogenase (LDH) (A020-2-1) were determined using commercial kits (Nanjing Jiancheng Institute of Bioengineering Research, Nanjing, China) according to the manufacturer’s description”.
Comment 3: Figure 2G and H: please add a higher magnification to evaluate tissue morphology.
Response 3: Thank you for raising this concern. In response to the reviewers’ feedback, we increased the magnification of HE-stained and Sirius red-stained images of liver tissues in Figures 2G and 2H, respectively, for assessing the tissue morphology.
Comment 4: Figure 5A: please add a higher magnification to evaluate tissue morphology.
Response 4: Thank you for pointing this out. We apologize for our unclear images. We had replaced the HE-stained image of the intestinal tissue with a larger magnification.
Comment 5: Western blot images: please add the molecular weights in western blot images.
Response 5: We apologize for not adding the corresponding molecular weight size of each protein when providing the western blotting image. We have now supplemented the information to ensure full and accurate. We appreciate your attention to this matter, and we hope that our revisions meet your expectations.
Comment 6: Please revise the punctuation and typing errors throughout the manuscript.
Response 6: We apologize for the typographical and punctuation errors in the article. We have now fully revised it to ensure full accuracy. We thank you for your interest in this matter and hope that our revisions will meet with your approval.
Comment 7: Figure 6: remove the error red lines.
Response 7: Thank you for bringing up this concern. The error red line in Figure 6 has been removed. We appreciate your attention to this matter, and we hope that our revisions meet your expectations.
Comment 8: The abbreviations must be written in full length the first time that they are mentioned.
Response 8: Thank you for the reminder. We apologize for using the abbreviations without writing the full length on the first mention. Based on the reviewers’ feedback, we have revised this section to improve it. We hope that our correction will be recognized by you.
We would like to express our sincere gratitude for the insightful comments and valuable feedback you provided on our manuscript. Your dedication to enhancing the quality of our paper has not gone unnoticed, and we are truly grateful for the time and effort you invested in this review process.
Your constructive critiques have taught us a great deal and have shed light on many areas where our manuscript can be improved. We have carefully considered each of your suggestions and have made every effort to address the identified shortcomings.
Once again, thank you for your diligent efforts. We have revised the manuscript to the best of our ability, and we hope that our revisions meet your expectations. Should you have any further questions or concerns, please do not hesitate to reach out to us.
Thank you once again for your invaluable contributions to our work.
Sincerely,
Tingtao Chen, Ph.D.
Professor, School of Pharmacy
Nanchang University
Nanchang 330031, China
Email: chentingtao1984@163.com or chentingtao@ncu.edu.cn
Reviewer 2 Report
Comments and Suggestions for Authors
Liver damage is now a critical public health problem. In their work, the authors focused on trying to eliminate toxic cadmium, which is one of the important factors resulting in liver damage by using the properties of the microorganisms that constitute the gut microbiota. This path of considerations and research seems to be extremely important and valuable, and the authors already have experience in this area. In the current work reviewed by me, the authors present the creation of a next-generation probiotic strain of E. coli 1917-pSK18a-MT (EM) capable of superficial expression of metallothionein, which is important in this aspect. The research was conducted on healthy mice, but the authors suggest using the promising and valuable nature of the obtained results in future studies in models of liver damage. In my opinion, the work was planned and prepared correctly. A rich and informative introduction provides a good introduction to the issue in question. The material and methods are described in detail, lists of primers and antibodies used are included in the supplement. The research results have a logical structure and are, in my opinion, complete, at the same time indicating the promising nature of the impact of the created strain, which undoubtedly requires further research in animal models with Cd-induced liver damage and others. I do not see any shortcomings in the presentation and description of the results. In my opinion, the discussion of the results is guided in the right direction, justifying at the same time the importance of the results obtained. The inference is correct. In my opinion, this is a very well-prepared, complete manuscript.
Author Response
Dear reviewer,
Thank you for reviewing our manuscript “The Next-Generation Probiotic E. coli 1917-pSK18a-MT Ameliorates Cadmium-Induced Liver Injury by Surface Display of Metallothionein and Modulation of Gut Microbiota” (Manuscript ID: nutrients-2976939), and we are truly grateful for the time and effort you invested in this review process. Thank you again for recognizing our work.
Sincerely,
Tingtao Chen, Ph.D.
Professor, School of Pharmacy
Nanchang University
Nanchang 330031, China
Email: chentingtao1984@163.com or chentingtao@ncu.edu.cn
Reviewer 3 Report
Comments and Suggestions for Authors
In the present manuscript, Zhang et al. generated a modified E. coli strain expressing metallothionein, as a probiotic to combat cadmium poisoning. The strain showed resistance to conditions mimicking the gastrointestinal tract and alleviated multiple symptoms associated with cadmium poisoning, such as hepatic fibrosis, inflammation, and oxidative stress, as well as gut dysbiosis and intestinal barrier damage. The research is compelling and clearly described in the manuscript. This work should be relevant to the field of novel strategies to overcome heavy meal poisoning in animals.
Minor remarks:
1. In the title, please consider replacing MT with its full name.
2. In line 96, please include the name and supplier of the ELISA kit.
3. In line 130, please correct "remover" to "removed".
4. In lines 117 and 197, please consider replacing "execution" with "sacrifice", which is a more commonly used word in animal research.
5. In lines 183-184 it reads: "(...) these strains were able to tolerate gastrointestinal environmental conditions". Please consider modifying this statement to "(...) these strains were able to tolerate conditions mimicking the gastrointestinal environment".
6. In section 2.5, please include the qPCR program (or its reference, in case a standard one was used).
7. In section 2.6, please identify the mass spectrometer system used and the acquisition parameters, if different from the ones in the cited reference.
8. In section 2.7, please change the first sentence to the past tense, to be consistent with the rest of the text. Please mention the percentage of polyacrylamide used in the SDS-PAGE gels.
9. In sections 2.8 and 2.9, please identify the kits used.
10. In the legend of Figure 1, please include the full names of the abbreviations used in this figure.
11. In section 3.5, please describe the histological features from Figure 5A in more detail
12. In Figure 5G, please consider typing the number in the central region in white, to see if the contrast relative to the background increases.
13. At the end of the discussion, please include the limitations of the study.
Comments on the Quality of English Language
Please see remarks #3 and #8.
Author Response
Dear reviewer,
Thank you for reviewing our manuscript “The Next-Generation Probiotic E. coli 1917-pSK18a-MT Ameliorates Cadmium-Induced Liver Injury by Surface Display of Metallothionein and Modulation of Gut Microbiota” (Manuscript ID: nutrients-2976939), and we appreciate the time and effort that you dedicated to providing feedback on our manuscript and are grateful for the insightful comments on and valuable improvements to our paper. The manuscript was carefully revised and point-by-point response was listed below. We hope that your comments have been addressed accurately. And those changes are highlighted in the manuscript by the revision pattern as well as in underline/red.
Response Comments:
Comment: In the present manuscript, Zhang et al. generated a modified E. coli strain expressing metallothionein, as a probiotic to combat cadmium poisoning. The strain showed resistance to conditions mimicking the gastrointestinal tract and alleviated multiple symptoms associated with cadmium poisoning, such as hepatic fibrosis, inflammation, and oxidative stress, as well as gut dysbiosis and intestinal barrier damage. The research is compelling and clearly described in the manuscript. This work should be relevant to the field of novel strategies to overcome heavy meal poisoning in animals.
Response: We thank the reviewer for appreciating our work and offering valuable insights for us to further improve our manuscript. The followings are our point-to-point responses to the minor remarks.
Minor remarks:
Comment 1: In the title, please consider replacing MT with its full name.
Response 1: Thank you for bringing this to our attention. We have changed the MT in the title to the full name.
Comment 2: In line 96, please include the name and supplier of the ELISA kit.
Response 2: We thank the reviewer for this comment. We have added the name and supplier of the ELISA kit in the updated manuscript. For your convenience, information on the kit is provided below for quick reference.
In lines 99-101, “(...) the expression of MT in EM was detected using a bacterial metallothionein enzyme-linked immunosorbent assay (ELISA) kit (Shanghai Tongwei Industry Co., Ltd.)”.
Comment 3: In line 130, please correct “remover” to “removed”.
Response 3: Thanks very much for this comment. In response to reviewer feedback, we have corrected the word “remover” to “removed”. We appreciate your attention to this matter, and we hope that our revisions meet your expectations.
Comment 4: In lines 117 and 197, please consider replacing “execution” with "sacrifice", which is a more commonly used word in animal research.
Response 4: We appreciate the reviewer’s insightful comment. We have replaced the word “execution” with the word “sacrifice”. We appreciate your attention to this matter, and we hope that our revisions meet your expectations.
Comment 5: In lines 183-184 it reads: “(...) these strains were able to tolerate gastrointestinal environmental conditions”. Please consider modifying this statement to “(...) these strains were able to tolerate conditions mimicking the gastrointestinal environment”.
Response 5: Thank you for bringing this out, we have changed the former lines 183-184 “(...) these strains were able to tolerate gastrointestinal environmental conditions” to now in lines 198-199 “(...) these strains were able to tolerate conditions mimicking the gastrointestinal environment”.
We are grateful for your guidance, which has helped us refine our manuscript, and we hope that these changes satisfactorily address the concerns raised.
Comment 6: In section 2.5, please include the qPCR program (or its reference, in case a standard one was used).
Response 6: Thank you so much for bringing this to our attention. We have now supplemented the qPCR amplification program in Section 2.5. For your convenience, the program settings for qPCR are provided below for quick reference.
In lines 138-140, “The qPCR amplification program was set to start at 95 ℃ for 10 min, followed by annealing at 95 ℃ for 30 s, then annealing at 60 ℃ for 30 s, and extending at 72 ℃ for 30 s for a total of 40 cycles”.
Comment 7: In section 2.6, please identify the mass spectrometer system used and the acquisition parameters, if different from the ones in the cited reference.
Response 7: Thank you for highlighting this aspect. We apologize for the lack of a comprehensive and detailed description of the mass spectrometer system and acquisition parameters used in Section 2.6 of the article. Recognizing the importance of strengthening this section, we have provided a detailed description of the section. We believe that these additions will strengthen the manuscript and provide the reader with a more comprehensive understanding of the subject. We have made revisions in Section 2.6:
“0.1 g of each group of mice feces and liver tissue samples were added to 1 mL of PBS for homogenization, and 3 mL of concentrated nitric acid was added and left overnight, after which they were transferred to a polytetrafluoroethylene ablation tube to be ablated at 80 ℃ for 1 h. At the end of the ablution, 2 mL of H2O2 solution was added to continue to heat up to 120 ℃ until it turned into a colorless and transparent liquid, and then cooled down to below 60 ℃, and fixed and mixed using ultrapure water. The Cd content in the samples was quantified by inductively coupled plasma emission spectrometry (ICP-MS) (CLeeman, Prodigy XP-High Dispersion, USA), and expressed as mg/g sample wet weight”.
Comment 8: In section 2.7, please change the first sentence to the past tense, to be consistent with the rest of the text. Please mention the percentage of polyacrylamide used in the SDS-PAGE gels.
Response 8: Thank you for your valuable suggestion. We apologize for the oversight of not changing the first sentence to the past tense and not adding the percentage of SDS-PAGE gels in Section 2.7. We have now changed these two errors. We appreciate your attention to this matter, and we hope that our revisions meet your expectations.
Comment 9: In sections 2.8 and 2.9, please identify the kits used.
Response 9: Thank you for your valuable feedback. We apologize for not accurately describing the kit information used in sections 2.8 and 2.9 of the article. To remedy this omission, we have added the following description to the article:
In section 2.8: “The levels of catalase (CAT) (A007-1-1), superoxide dismutase (SOD) (A001-1-1), malondialdehyde (MDA) (A003-1-1) in liver tissue, as well as the serum levels of alanine aminotransferase (ALT) (C009-2-1), aspartate aminotransferase (AST) (C010-2-1), alkaline phosphatase (ALP) (A059-2-1) and lactate dehydrogenase (LDH) (A020-2-1) were determined using commercial kits (Nanjing Jiancheng Institute of Bioengineering Research, Nanjing, China) according to the manufacturer's description”.
In section 2.9: “Microbial genomic DNA was extracted using a DNA extraction kit (Tiangen, Beijing, China, DP712) according to the manufacturer's instructions (......)”.
We hope that our correction will be recognized by you.
Comment 10: In the legend of Figure 1, please include the full names of the abbreviations used in this figure.
Response 10: We are grateful for the reviewer’s comment. We apologize that the full names of the abbreviations used in this figure are not included in the legend of Figure 1. In the revised manuscript, we have adjusted the description of this section as follows,
In lines 208-209, “(...) (F) Antioxidant properties of E and EM (n = 3). E, E. coli 1917; EM, E. coli 1917-pSK18a-MT; MT, metallothionein; h, hour; d, day”.
Comment 11: In section 3.5, please describe the histological features from Figure 5A in more detail.
Response 11: Thank you so much for bringing this to our attention. We apologize for not describing the histological features in Figure 5A in Section 3.5 in detail. In the revised manuscript, we have adjusted the description of this section as follows,
In lines 301-305, “HE staining showed significant inflammatory infiltration and intestinal epithelial damage in colonic tissues of group M as compared to group C, altering the morphological structure of the intestinal tract to a certain extent and showing significant lesions, which was interestingly attenuated more dramatically by EM treatment than in group E (Figure 5A)”.
Comment 12: In Figure 5G, please consider typing the number in the central region in white, to see if the contrast relative to the background increases.
Response 12: Thank you so much for bringing this to our attention. We apologize for the confusion caused by the unclear picture. As per the reviewer’s comment, we have placed a more obvious number in the center region of Figure 5G for your quick reference.
Comment 13: At the end of the discussion, please include the limitations of the study.
Response 13: Thank you for raising this question. Indeed, there are limitations to this study, and we have included them at “Conclusions” to better reflect the rigor of the article.
In lines 454-458, “However, since this study only explored the mechanism of action of E. coli 1917-pSK18a-MT on foodborne Cd exposure-induced liver injury in the healthy male mice model, as well as the uncertainty of the expression of engineered bacteria in affecting gut-associated metabolites in the mice, which had a non-negligible impact on the conclusions that we drew”.
Thank you once again for your valuable feedback. This will greatly enhance the completeness of our manuscript. We hope our correction will be recognized by you.
I would like to take this opportunity to thank you for all your time involved and this great opportunity for us to improve the manuscript. I hope you will find this revised version satisfying.
Sincerely,
Tingtao Chen, Ph.D.
Professor, School of Pharmacy
Nanchang University
Nanchang 330031, China
Email: chentingtao1984@163.com or chentingtao@ncu.edu.cn